

# Validating anthropogenic threat maps as a tool for assessing river ecological integrity in Andean–Amazon basins

Janeth Lessmann[1,2,3], Maria J. Troya[1], Alexander S. Flecker[4],
W. Chris Funk[5], Juan M. Guayasamin[1,6], Valeria Ochoa-Herrera[7,8],
N. LeRoy Poff[5,9], Esteban Suárez[1] and Andrea C. Encalada[1,10,11]

[1] Instituto BIÓSFERA, Colegio de Ciencias Biológicas y Ambientales, Universidad San Francisco de Quito, Quito, Ecuador
[2] Departamento de Ecología, Pontificia Universidad Católica de Chile, Santiago de Chile, Chile
[3] Instituto de Ecología y Biodiversidad, Santiago de Chile, Chile
[4] Department of Ecology & Evolutionary Biology, Cornell University, Ithaca, NY, USA
[5] Department of Biology, Graduate Degree Program in Ecology, Colorado State University, Fort Collins, CO, USA
[6] Centro de Investigación de la Biodiversidad y Cambio Climático (BioCamb) e Ingeniería en Biodiversidad y Recursos Genéticos, Facultad de Ciencias de Medio Ambiente, Universidad Tecnológica Indoamérica, Quito, Ecuador
[7] El Politécnico, Colegio de Ciencias e Ingenierías, Universidad San Francisco de Quito, Quito, Ecuador
[8] Department of Environmental Sciences and Engineering, Gillings School of Global Public Health, University of North Carolina at Chapel Hill, Chapel Hill, NC, USA
[9] Institute for Applied Ecology, University of Canberra, Canberra, ACT, Australia
[10] MARE, Department of Life Sciences, Universidade de Coimbra, Coimbra, Portugal
[11] Department of Geography, University of North Carolina at Chapel Hill, Chapel Hill, NC, USA

Corresponding author
Janeth Lessmann, jdlessmann@uc.cl

## ABSTRACT

Anthropogenic threat maps are commonly used as a surrogate for the ecological integrity of rivers in freshwater conservation, but a clearer understanding of their relationships is required to develop proper management plans at large scales. Here, we developed and validated empirical models that link the ecological integrity of rivers to threat maps in a large, heterogeneous and biodiverse Andean–Amazon watershed. Through fieldwork, we recorded data on aquatic invertebrate community composition, habitat quality, and physical-chemical parameters to calculate the ecological integrity of 140 streams/rivers across the basin. Simultaneously, we generated maps that describe the location, extent, and magnitude of impact of nine anthropogenic threats to freshwater systems in the basin. Through seven-fold cross-validation procedure, we found that regression models based on anthropogenic threats alone have limited power for predicting the ecological integrity of rivers. However, the prediction accuracy improved when environmental predictors (slope and elevation) were included, and more so when the predictions were carried out at a coarser scale, such as microbasins. Moreover, anthropogenic threats that amplify the incidence of other pressures (roads, human settlements and oil activities) are the most relevant predictors of ecological integrity. We concluded that threat maps can offer an overall picture of the ecological integrity pattern of the basin, becoming a useful tool for broad-scale conservation planning for freshwater ecosystems. While it is always advisable to have finer scale in situ measurements of

ecological integrity, our study shows that threat maps provide fast and cost-effective results, which so often are needed for pressing management and conservation actions.

## INTRODUCTION

Effective management of large landscape units is an emerging priority for freshwater biodiversity conservation. Several ecosystem processes, the distribution of many species, and the threats they experience occur throughout large geographical areas that often encompass whole watersheds (*Collares-Pereira & Cowx, 2004*; *Thieme et al., 2007*). Moreover, the ecological integrity of stream and river ecosystems depends on the connectivity they provide along their courses and across the terrestrial ecosystems with which they interact (*Higgins et al., 2005*). Therefore, when conservation efforts are conducted at a large watershed scale, maps that represent the spatial variation of the ecological integrity of freshwater systems are a useful tool to allocate and prioritize conservation resources, and to inform planning for monitoring, and ecosystem-based management (*Abell et al., 2002*; *Carlisle, Falcone & Meador, 2009*; *Clapcott et al., 2011*). However, mapping the ecological integrity for these large geographical scales requires extensive and costly fieldwork, hampering our ability to generate the information needed for management (*Abell et al., 2002*; *Revenga et al., 2005*). As a result, freshwater ecosystems are often underrepresented in land-use planning and conservation initiatives along large watersheds (*Revenga et al., 2005*; *Comer & Faber-Langendoen, 2013*). Here, we develop and test a geographical model to depict the ecological integrity of a large tropical watershed, based on the distribution of human activities at the landscape level.

As an alternative to extensive field assessments of ecological integrity, geographic information systems (GIS) and remote sensing technology can generate and synthesize spatial data on the distribution of human activities (stressors to freshwater ecosystems), that is geographically explicit, and easier to assess and monitor throughout large watersheds (*Revenga et al., 2005*; *Nel et al., 2009*). Thus, maps that represent the location and intensity of such stressors (i.e., anthropogenic threat maps) are used in conservation as surrogates for the system's ecological integrity or to make inferences about it (*Nel et al., 2009*; *Linke, Turak & Nel, 2011*). For instance, maps on agriculture land use and location of hydroelectric power plants have been used to infer the impairment of ecological integrity in a watershed or adjacent rivers (*Thieme et al., 2007*; *Esselman & Allan, 2011*; *Finer & Jenkins, 2012*; *Winemiller et al., 2016*; *Latrubesse et al., 2017*).

Despite the advantages of using anthropogenic threats maps as surrogates of ecological integrity, proper assessments of freshwater ecosystems are critically needed to understand basin-wide threat patterns, interactions, and scales of response of ecological integrity (*Mattson & Angermeier, 2007*; *Tulloch et al., 2015*). There is high uncertainty regarding the use of coarse and relatively static spatial information on anthropogenic threats to estimate

levels of ecological integrity, which is more influenced by local and temporal changes (*Clapcott et al., 2011*). In addition, important threats to rivers such as discharges of domestic and industrial effluent or the introduction of exotic species, are impossible to detect with remote imagery (*O'Neill et al., 1997*). Therefore, the field of freshwater conservation needs evaluations to determine if planning based on anthropogenic threat maps, as an indicator of ecological integrity, are reliable and reflect the actual conservation needs of rivers (*Clavero et al., 2010*). Specifically, we need to develop empirical models to determine the explanatory power of anthropogenic threat maps for freshwater ecological integrity (*Carlisle, Falcone & Meador, 2009*). Moreover, validating these models with in situ measurements is critical to assess the robustness of anthropogenic threat maps in predicting the conditions of unsampled areas (*Carlisle, Falcone & Meador, 2009*).

According to previous studies, the ability of geospatial data on threats to predict ecological integrity can be highly variable (*Amis et al., 2007*; *Carlisle, Falcone & Meador, 2009*; *Clapcott et al., 2011*; *Thornbrugh et al., 2018*). Moreover, it is not clear to what extent the accuracy of these predictions is influenced by the set of predictors used in the modeling, the spatial scale of the analysis, and the natural and social context of each study area. For example, land use variables, such as agriculture and urbanization, have been considered crucial predictors due to their considerable impact in regulating stream processes (*Allan, 2004*; *Nel et al., 2009*). However, considering the high geographic heterogeneity of some regions, the addition of environmental variables and their covariation with anthropogenic gradients could significantly improve the prediction of the ecological integrity (*Allan, 2004*). There is also an increasing interest in the spatial scale at which land use influences stream ecosystem health (*Sheldon et al., 2012*). Rivers are hierarchically organized systems, in which large-scale characteristics and presence of human threats influence the ecological integrity of watercourses (*Oliveira & Cortes, 2006*). Thus, the power to predict the ecological integrity for regional-scale units, such as microbasins (i.e., the entire drainage area of a river), might be higher than for local scales (i.e., individual river sites). Finally, for regions with multiple stressors and heterogeneous freshwater systems, the links between ecological integrity and stressors are expected to be difficult to depict (*Leal et al., 2016*; *Schinegger et al., 2016*). In this context, the ability of threat maps to infer the health of rivers could be limited. Therefore, studies are needed to assess and validate the circumstances for which predictive models based on anthropogenic threat maps are a useful tool to estimate freshwater ecological integrity and inform management objectives.

The large watersheds that originate in the Andes and give rise to the main tributaries that form the Amazon river are outstanding examples of the ecological and socio-economic complexities that condition the management of large and heterogeneous landscape units. These basins harbor astonishing levels of freshwater species diversity while providing important services to local populations (*Collen et al., 2014*). At the same time, human activities such as hydropower development, agricultural expansion, and oil industry are encroaching into these basins, causing large-scale degradation of freshwater ecosystems (*Finer & Jenkins, 2012*; *Finer et al., 2015*; *Latrubesse et al., 2017*). Given the intimate ecological and hydrological connections between the Andean

mountains and Amazonian rivers (*Encalada et al., 2019b*), proper management and mitigation of these threats require basin-wide monitoring and integrative strategies across their large elevation gradients (*Castello & Macedo, 2016*). However, the development of such integrative approaches is hindered by the remoteness that characterize these watersheds, their landscape heterogeneity, and the lack of funds to implement ecological monitoring at a large spatial scale (but see *Winemiller et al., 2016*; *Latrubesse et al., 2017*). In this context, the development and validation of geographical models depicting the extent, distribution, and impact of anthropogenic threats offers an opportunity to inform future management initiatives at these critical watersheds.

In this study, we evaluated the usefulness of an anthropogenic threat map as an index of the ecological integrity of rivers in a key Amazon–Andean watershed: the upper Napo River basin in Ecuador. Specifically, we ask: (1) How well do anthropogenic threat maps predict direct measures of river ecological integrity? (2) Which anthropogenic threats and environmental characteristics best explain variation in ecological integrity? (3) How do predictions differ across spatial scales? For this, we first estimated, via in situ fieldwork, the ecological integrity in several rivers throughout the altitudinal gradient of the basin. Then, we mapped the location, extent, and magnitude of impact of different anthropogenic threats to freshwater ecosystems using geospatial data. Finally, we developed and validated empirical models that link the condition of rivers to spatial information of anthropogenic threats. We also studied the influence of the scale of the analysis unit (local and regional), and the use of environmental variables in the predictions. While assessing this tool, our work offers recommendations about the use of anthropogenic threat maps for conservation planning in other Andean–Amazon watersheds, which urgently need expeditious and reliable strategies for management and conservation actions.

## MATERIALS AND METHODS

### Study area

The Napo River basin encompasses an extensive gradient from the summits of the Andes, to the lowlands of the Amazon. Thus, this basin exhibits exceptional diversity of freshwater species and ecosystems, which competes with an expanding influence of urban settlements, intensive agriculture, oil, mining, and hydroelectric projects (*Anderson & Maldonado-Ocampo, 2011*; *Lessmann et al., 2016a*, *2016b*). These changes occur in a context in which effective management is hampered by lack of funds and reliable information to detect ongoing ecosystem degradation trends.

In this study, we focused on the 59,573 km$^2$ corresponding to the upper part of the Napo basin, in Ecuador (Fig. 1). The section of the basin we studied ranges from 5,897 m in the Andes, to 200 m in the Eastern lowlands. The highest slopes of the basin are dominated by páramo ecosystems (i.e., humid mountain steppe composed of shrubs and grasses) and upper montane forests, covering a complex topography and steep slopes that give rise to cold, fast-flowing streams. As they descend into the lower montane forests (1,600–600 m) and approach the Amazonian lowlands (<600 m), streams become warmer, and wider, eventually turning into large, slow-moving and sediment-loaded lowland rivers.

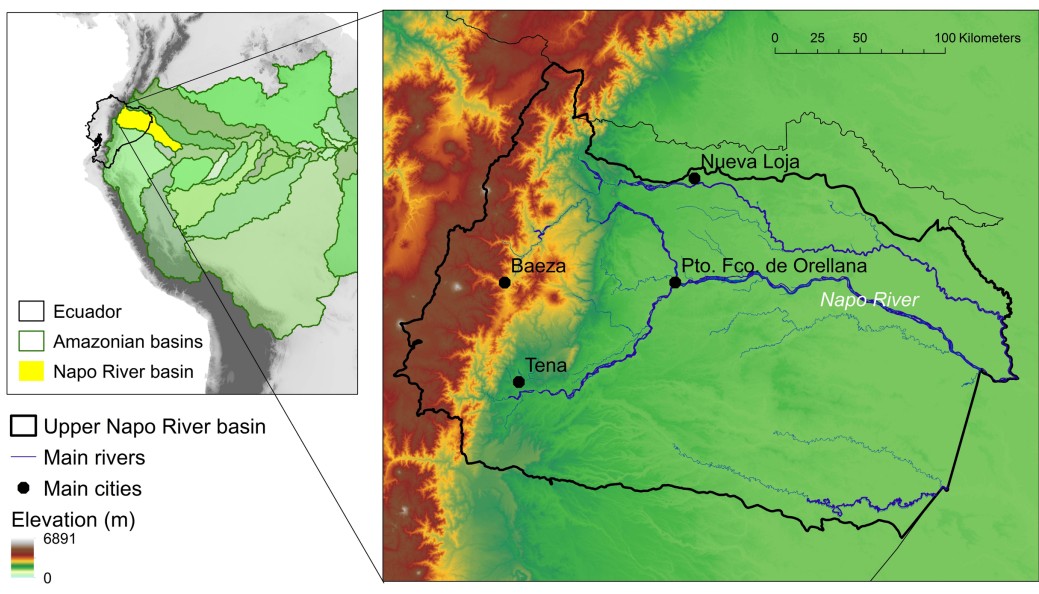

**Figure 1 Study area.** The upper Napo river basin is an Andean–Amazon watershed in Ecuador with a large altitudinal gradient and heterogeneous freshwater systems.

The basin also presents a gradient in human pressures; whereas the eastern Amazonian lowlands contain well conserved ecosystems, the north-central area suffers from the impact of rapid expansion of extractive and agricultural activities and urban development.

## Ecological integrity assessment

To characterize ecological integrity along the upper Napo basin, we established 140 sampling sites (stream width 2–20 m) across the altitudinal gradient, from 231 to 3915 m (Fig. 2). In all cases, we selected sites to represent the heterogeneous characteristics of rivers and anthropogenic pressures found in the area. Sampling was performed between 2011 and 2014. Field permits were granted by The Ecuadorian Ministry of Environment (Permits: #56-IC-FAU/FLO-DPN/MA, MAE-DNB-CM-2015-0017).

Ecological integrity is defined as "the degree to which the physical, chemical and biological components (including composition, structure, and process) of an ecosystem and their relationships are present, functioning and maintained close to a minimally impacted reference condition" (*Schallenberg et al., 2001*). Although ecological integrity is a widely used concept, the parameters with which it is characterized vary from one ecosystem to another, especially when it comes to indicator species (*Carignan & Villard, 2002*). In this study, we assessed the ecological integrity of rivers through three components (Appendix A): (i) stream biotic community composition, (ii) habitat integrity, and (iii) water physical-chemical characteristics.

To assess the biotic community composition, at each stream/river site we sampled benthic aquatic invertebrates using Surber samples (five replicates) and kick samples (one per stream; as in (*Gill et al., 2016*)). In the laboratory, samples were sorted and classified to the lowest possible taxonomic level following (*Domínguez & Fernández, 2009*).

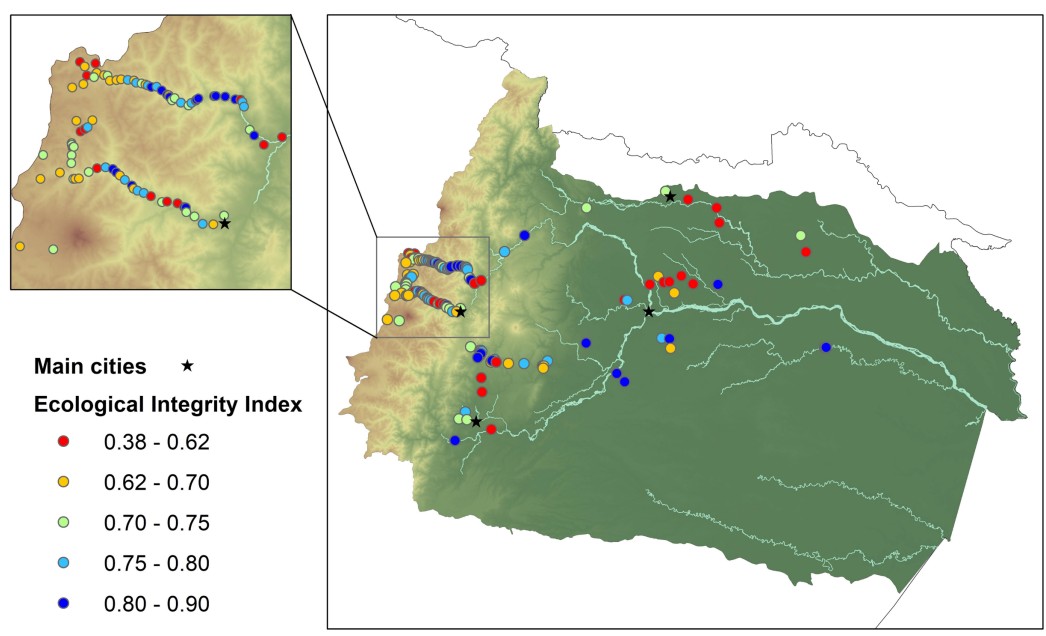

**Figure 2  Ecological integrity index (EII) for all sampled rivers in the upper Napo river basin.** Low values of EII (red color) indicate rivers with the poorest ecological integrity, whereas rivers with EII values near one (dark blue color) indicate the highest integrity.

Then, each identified sample was rated according to its level of physiological tolerance, using the Amazon Biotic Index (AMBI) (*Encalada et al., 2019a*), which considers a wide pool of invertebrate families from the Andean–Amazon gradient. We also calculated the Average Score Per Taxa (ASPT) (*Roldán-Pérez, 1999*), which is a parameter that standardizes the AMBI values by the total number of families found in each stream site and represents the biological integrity based on the identity of such families.

  We evaluated the habitat quality along each stream/river site through the Riparian Quality Index (QBR-Am) and Fluvial Habitat Index (IHF-Am) (*Encalada et al., 2019a*). The QBR-Am index evaluates the overall condition of riparian habitat by synthesizing several variables including structure, quality, and species composition of riparian vegetation coverage. On the other hand, IHF-Am incorporates physical and morphological variables such as stream velocity, depth, frequency of riffles/pools/glides, presence of meanders, substrate diversity and composition, and primary producer composition (*Pardo et al., 2002*). Both indices were modified taking into account the heterogeneous nature of the upper Napo basin (*Encalada et al., 2019a*).

  To evaluate water physical-chemical characteristics, we sampled four key parameters at each river (pH, conductivity, dissolved oxygen, and temperature). Then, we estimated the deviations from human development by calculating the absolute difference between observed values and references values. These reference values were obtained from *Alexiades et al. (2019)*.

  We scaled all parameters (AMBI and ASPT, QBR-Am, IHF-Am, and physical-chemical measurements) between zero and one, with the highest value indicating the highest possible level of ecological integrity for that parameter. These parameters we averaged

within their corresponding component. The final step in producing an index was combining the three components into a single value that represents the overall integrity. Here, we used the arithmetic mean because we were interested in producing a straightforward and understandable index, rather than more sophisticated methods that lose less information but are more unintelligible for conservation planners (*Andreasen et al., 2001*). Thus, for each river sampled, the Ecological Integrity Index (EII) was obtained by averaging the values of the three components. EII values of zero indicate the poorest ecological integrity, while values of one indicate the highest integrity. We also built an additional metric to contrast the results of the analysis obtained from the arithmetic mean. The biological component can provide a comprehensive measure of ecological integrity because aquatic organisms reflect the cumulative effects of environmental disturbances and pollution (*Oliveira & Cortes, 2006*). Therefore, for the second metric, we gave more weight to the biological component in comparison to the other components (2:1:1) (*Andreasen et al., 2001*).

## Anthropogenic threat maps to freshwater ecosystems

We generated anthropogenic threat maps for freshwater ecosystems of the upper Napo basin at ~0.25 km$^2$ resolution in ArcMap 9.3 (Esri, Redlands, CA, USA). Specifically, we evaluated the following threats: human settlements (villages and cities), mining and quarrying (gold, lead), oil activities (wells, oil spills, oil pools and pipelines), hydroelectric and thermoelectric power plants, agricultural land use, water withdrawals, aquaculture farms, and roads (Table 1). These threats have been linked to the ecological integrity of rivers. For example, agriculture, mining, and human settlements are used to infer information on water use, sedimentation, chemical, and nutrient pollution; hydroelectric power and roads can produce hydrologic alterations and fragmentation, and aquaculture farms are related to water pollution and invasive species (*Stein, Stein & Nix, 2002*; *Nel et al., 2009*). From different Ecuadorian institutions, we gathered digital information on the location and categories that comprise each threat (e.g., types of crops, level of transit on roads, number of inhabitants in cities or villages). This spatial information was contemporaneous (2012–2013) with the data on ecological integrity collected at the rivers.

We built individual maps for each threat representing its location, extent, and magnitude of impact on the ecological integrity of freshwater systems (*McPherson et al., 2008*). By consulting specialized references, we assigned a magnitude of impact (from zero to one) to the different categories within each threat (Table 1). For example, in the case of human settlements, the category of urban areas received a relatively higher impact value (0.65) than villages (0.35). Additionally, we assigned a maximum distance (i.e., buffer) at which a threat category has a negative impact on freshwater ecological integrity, with this impact diminishing linearly within the buffer (Table 1). When neighboring buffers overlapped, we added the magnitude of their impact. As a result, the nine anthropogenic threats maps summarize the relative magnitude of their impact scaled between zero and one, where one represent the maximum level of impact of a threat on freshwater systems.

**Table 1 Values of the magnitude of impact and the impact distance used to construct anthropogenic threat maps for freshwater systems in the upper Napo river basin.** We collected spatial data on human threats directly from government ministries and NGOs following official information requests. The assignment of magnitude of impact and distance values was a logical process informed by the literature and our knowledge of the study area.

| Anthropogenic threat | Categories within threats | | Magnitude of impact | Impact distance | Data sources | References used for assigning the magnitude and distance of impact |
|---|---|---|---|---|---|---|
| Human settlements (scaled between 0–1 according to population density) | Urban areas | | 0.65 | 10 km | Instituto Geográfico Militar. 2012. Base Regional, 1:250.000 (http://www.igm.gob.ec/). Instituto Nacional de Estadística y Censos. Densidad poblacional. 2012. (http://www.ecuadorencifras.gob.ec/) | Magnitude: *Stein, Stein & Nix (2002)*; *McPherson et al. (2008)*; *Esselman & Allan (2011)* Distance: *McPherson et al. (2008)*; *Esselman & Allan (2011)* |
| | Village | | 0.35 | 3 km | | |
| Mining | Construction material mining | | 0.6 | 5 km | Agencia de Regulación y Control Minero. 2012. Mapa del Catastro Minero Nivel Nacional. 1:1.400.000 (http://www.controlminero.gob.ec/) The Nature Conservancy of Ecuador. 2013 | Magnitude: *Tiwary (2001)*; *Stein, Stein & Nix (2002)*; *McPherson et al. (2008)*; *Romero et al. (2008)* Distance: *Stein, Stein & Nix (2002)*; *McPherson et al. (2008)* |
| | Metal mining | | 0.4 | | | |
| | Non-metal mining | | 0.1 | | | |
| Agricultural land use | Permanent, semi-permanent, and annual crops | | 0.6 | 5 km | Socio Bosques. Estimación de la Tasa de Deforestación del Ecuador continental. 2012. (http://sociobosque.ambiente.gob.ec/) | Magnitude: *Del Tánago (1996)*; *Stein, Stein & Nix (2002)*; *McPherson et al. (2008)* Distance: *McPherson et al. (2008)* |
| | Agricultural mosaic and pastureland | | 0.4 | | | |
| Hydroelectric power plants (size based on generated power) | Operating | Large size | 0.375 | Scaled to plant size (max. 30 km) | Agencia de Regulación y Control de Electricidad. 2012. (http://www.regulacionelectrica.gob.ec/) The Nature Conservancy of Ecuador. 2013 | Magnitude: *Stein, Stein & Nix (2002)*; *Finer & Jenkins (2012)* Distance: *McPherson et al. (2008)* |
| | | Medium size | 0.225 | | | |
| | | Small size | 0.15 | | | |
| | Under construction | Large size | 0.125 | | | |
| | | Medium size | 0.075 | | | |
| | | Small size | 0.05 | | | |
| Thermoelectric power plants | Without categories | | 1 | 2 km | Agencia de Regulación y Control de Electricidad. 2012. (http://www.regulacionelectrica.gob.ec/) | Distance: adapted from *Stein, Stein & Nix (2002)*; *Verones et al. (2010)* |
| Oil activities | Wells | Operating | 0.28 | 1.5 km | Sistema Nacional de Información de la Reparación Integral. Mapa de afectaciones ambientales. 2012. (http://pras.ambiente.gob.ec/siesap), The Nature Conservancy of Ecuador. 2013 | Magnitude: *Rosenfeld, Gordon & Guerin-McManus (1997)*; *Stein, Stein & Nix (2002)*; *O'Rourke & Connolly (2003)*; *San Sebastian & Hurtig (2004)* Distance: *Stein, Stein & Nix (2002)* |
| | | No operating | 0.07 | | | |
| | Recent oil spills | High volume | 0.15 | 5 km | | |
| | | Low volume | 0.09 | | | |
| | Old oil spills | High volume | 0.075 | | | |
| | | Low volume | 0.015 | | | |
| | Oil pools | Without treatment | 0.2 | | | |
| | | Recent with treatment | 0.0625 | | | |
| | | Old without treatment | 0.0125 | | | |
| | Pipelines | In risky areas (e.g., landslides) | 0.07 | 30 m | | |
| | | Out of risk areas | 0.03 | | | |

| Table 1 (continued). | | | | | |
| --- | --- | --- | --- | --- | --- |
| Anthropogenic threat | Categories within threats | Magnitude of impact | Impact distance | Data sources | References used for assigning the magnitude and distance of impact |
| Aquaculture farms | Without categories | Scaled from 0.1 to 1 according to water volume | 1 km | Secretaría del Agua. 2013. (http://www.agua.gob.ec/) | Distance: *Buschmann (2001)*; *Stein, Stein & Nix (2002)* |
| Water withdrawals | Without categories | Scaled from 0.1 to 1 according to water volume | 1 km | Secretaría del Agua. 2013. (http://www.agua.gob.ec/) | Distance: adapted from *McPherson et al. (2008)* |
| Roads | Primary road | 0.5 | 1 km | Instituto Geográfico Militar. 2012. Base Regional, 1:250.000 (http://www.igm.gob.ec/) | Magnitude: *Stein, Stein & Nix (2002)*; *McPherson et al. (2008)*; *Esselman & Allan (2011)* Distance: *Stein, Stein & Nix (2002)*; *Celi (2005)* |
| | Secondary road | 0.3 | | | |
| | Local road | 0.2 | | | |

## Ecological integrity predictive models

As a first statistical analysis, we executed a Moran Test I (Esri, Redlands, CA, USA) to evaluate if ecological integrity levels of sites that lie close to each other were independent. The test indicated that the observed pattern of ecological integrity is consistent with a random distribution ($Z$ score 1.55), obviating the need to use statistical methods that deal with spatial autocorrelation (*Fortin & Dale, 2005*).

Through statistical models (see details below), we estimated the relative importance of each anthropogenic threat as a predictor of the ecological integrity of rivers. Moreover, we tested if the accuracy of these predictions was influenced by the scale of the analysis unit (local vs. regional), and by the inclusion of environmental variables critical for lotic ecosystems. Environmental variables included elevation, slope, water flow accumulation (derived from a digital elevation model, http://srtm.csi.cgiar.org/), and precipitation (from Ministerio del Ambiente del Ecuador 2013). Based on this information, we developed four different modeling scenarios: (a) prediction of local ecological integrity from anthropogenic threat variables alone, (b) prediction of local ecological integrity from anthropogenic threat and environmental variables, (c) prediction of regional ecological integrity from anthropogenic threat variables alone, and (d) prediction of regional ecological integrity from anthropogenic threat and environmental variables. For modeling scenarios at local scales (a and b), we used the same unit of analysis as for the anthropogenic threat maps (pixels of ~0.25 km$^2$), and we assigned the EII value of each sampled river to the pixel where it occurs. In the case of the modeling scenarios (c) and (d), we classified the basin into 245 microbasins (average size = 243 km$^2$, SD = 210), which were derived from a 250 m digital elevation model, through ARC Hydro tool (Esri, Redlands, CA, USA). Values of EII, anthropogenic threats, and environmental variables within a microbasin were averaged, resulting in 29 sampled microbasins in the study area.

For each modeling scenario, we fit generalized linear models (GLMs, Gaussian distribution) in R v. 3.1.3 (*R Core Team, 2014*). Using the dredge function implemented in the MuMIn package (*Barton, 2009*), we started with a global model that contained all explanatory variables, followed by sub-model sets from the global model. In addition, we limited the maximum number of terms in the sub-models to seven in the case of the model scenarios a and b ($N = 140$), and two for model scenarios c and d ($N = 29$). We selected as the best model the one with lowest AICc (Akaike's information criterion corrected for small sample sizes), excluding models containing collinear predictors (those with correlation Pearson coefficient higher than 0.7) (*Burnham & Anderson, 2002*). The best models selected from the modeling scenarios were used to generate maps that show the predicted ecological integrity at local and regional scales across the entire upper Napo River basin.

To obtain a more reliable estimate of the prediction accuracy of ecological integrity, models should be validated with data that are independent of the data used to build the model (*Fielding & Bell, 1997*). However, as we lack independent data, we used the *k*-fold cross-validation procedure. In comparison to other methods, the cross validation allows an efficient and unbiased estimation of the predictive power of models by taking average results from *k* partitions of the data (*Kuhn & Johnson, 2013*). We used the Caret package (*Kuhn, 2008*) to randomly split the data into seven groups of equal size. Then, coefficients of the predictors from the best model were adjusted using the data of the first six groups (85% of observations). The prediction performance ($R^2$) of the adjusted model was estimated by comparing the predicted values with the observed values from the seventh group (the 15% of observations not used to adjust the model). We repeated this procedure 20 times with each group as an evaluation set and the remaining six as a fitting set. The overall predictive performance of the best model was estimated through the mean of all coefficients of determination ($R^2$) values from the 20 repetitions. Appendix B describes the R routine used to build the predictive models.

# RESULTS

## Sampled ecological integrity and anthropogenic threat maps

Rivers of the upper Napo basin displayed values of ecological integrity ranging from 0.38 to 0.90 (Fig. 2; Appendix C). Although no spatial pattern of ecological integrity was obvious, rivers at higher elevation had the highest number of sites with good ecological integrity (>0.75 EII). In contrast, rivers with the lowest ecological integrity (<0.62 EII) were mostly located nearby large cities. Moreover, ecological integrity values obtained from the weighted metric (Appendix C) were, in general, lower than those from the arithmetic mean (mean of differences = 0.05, $t = -12.1$, $p < 0.0001$) but displayed a similar spatial pattern throughout the basin ($r = 0.88$, $p < 0.0001$).

According to the anthropogenic threat maps (Fig. 3), threats with the greatest extent in the basin were agricultural land use (52% of the basin), human settlements (21%), mining (15%), and oil activities (10%). In contrast, the impact of power plants, water withdrawals, and aquaculture farms was more spatially limited. Furthermore, we observed
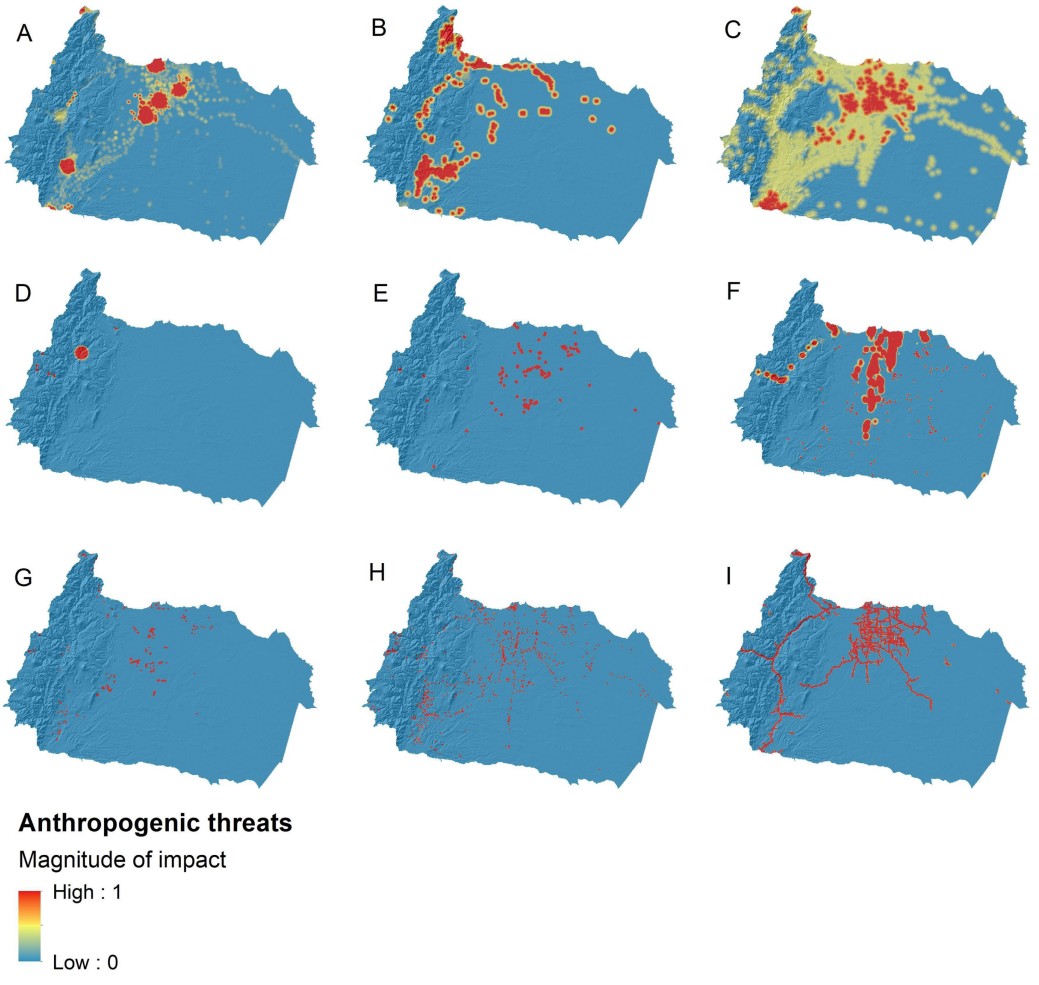

**Anthropogenic threats**

Magnitude of impact

High : 1

Low : 0

**Figure 3 Anthropogenic threat maps for freshwater ecosystems in the upper Napo river basin.** The magnitude of impact of each threat ranges between zero and one. Values closer to one are sites where a given threat has the maximum impact to freshwater systems. In contrast, an impact value of zero represents sites with no records of such pressures. Evaluated anthropogenic threats were: (A) human settlements, (B) mining, (C) agricultural land use, (D) hydroelectric power plants, (E) thermoelectric power plants, (F) oil activities, (G) water withdrawals, (H) aquaculture farms, and (I) roads.

that several anthropogenic features (agricultural land use, mining, oil activities, aquaculture farms) were restricted to areas near roads, especially in the lowlands.

## Predictive models of ecological integrity

Based on the cross-validation analysis (Table 2), we found that the best models from each modeling scenarios differed in their predictive ability. There was limited power to predict ecological integrity at a local scale using only anthropogenic threat variables as predictors (model a: mean $R^2$ = 0.16, SD = 0.14). However, the prediction accuracy of modeling at a local scale improved with the inclusion of environmental variables (model b: mean $R^2$ = 0.23, SD = 0.16), and more so if we carried out the analyses at a regional scale (model c and d: mean $R^2$ = 0.47, SD = 0.32). Notice that environmental variables were

**Table 2 Best models for predicting the ecological integrity index (EII) in four different modeling scenarios (a–d).**

| Modeling scenarios | Selected variables | Coefficients | Mean $R^2$ (from cross-validation) |
|---|---|---|---|
| a. River EII prediction, with threat variables ($N = 140$) | *Intercept* | 0.738*** | 0.16 (SD 0.14) |
| | Human settlements | −0.145* | |
| | Roads | −0.080 | |
| | Oil activities | −0.106 | |
| b. River EII prediction, with threat and environmental variables ($N = 140$) | *Intercept* | 0.762*** | 0.23 (SD 0.16) |
| | Human settlements | −0.165* | |
| | Roads | −0.121** | |
| | Oil activities | −0.136* | |
| | Elevation | −0.176*** | |
| | Slope | 0.199*** | |
| c. Microbasin EII prediction with threat variables ($N = 29$) | *Intercept* | 0.767*** | 0.47 (SD 0.32) |
| | Roads | −1.576*** | |
| d. Microbasin EII prediction with threat and environmental variables ($N = 29$) | *Intercept* | 0.767*** | 0.47 (SD 0.32) |
| | Roads | −1.576*** | |

**Note:**
All predictor variables, included environmental factors, were scaled between 0 and 1.
* $p < 0.05$.
** $p < 0.01$.
*** $p < 0.001$.

not part of the best models at the regional scale, and therefore the results of modeling scenarios c and d are the same.

Regarding the predictor variables, roads and human settlements were often selected in the best models (and several sub-optimal models), indicating that rivers near roads, cities, and villages have lower ecological integrity (Table 2). The presence of oil industry activities was also associated with lower levels of ecological integrity at a local scale. Slope and elevation were also selected at a local scale, having a positive and negative relationship with ecological integrity, respectively. Sub-optimal models are available in Appendix D. Best models based on the EII constructed with weighted metric has slightly lower predictive powers than models based on the arithmetic mean. However, models for both metrics showed similar relationships between ecological integrity and predictors (Appendix D).

Although the modeling scenarios differed in terms of predictive power, predictor variables, and spatial scales, the general spatial patterns of predicted ecological integrity was similar (Fig. 4): (1) main roads and large cities predicted poor river conditions (<0.62 EII), (2) remote areas of Amazonian lowlands, which have little presence of anthropogenic threats were projected to have high integrity (>0.70 EII), and (3) when environmental variables were included, rivers in the Andean foothills were predicted to have a very high ecological integrity (>0.80 EII).

## DISCUSSION

Understanding the geographical pattern of the anthropogenic threats and their spatial relationships with ecological integrity of rivers is a basic but often neglected aspect of freshwater conservation planning (*Clavero et al., 2010*). Since anthropogenic threat

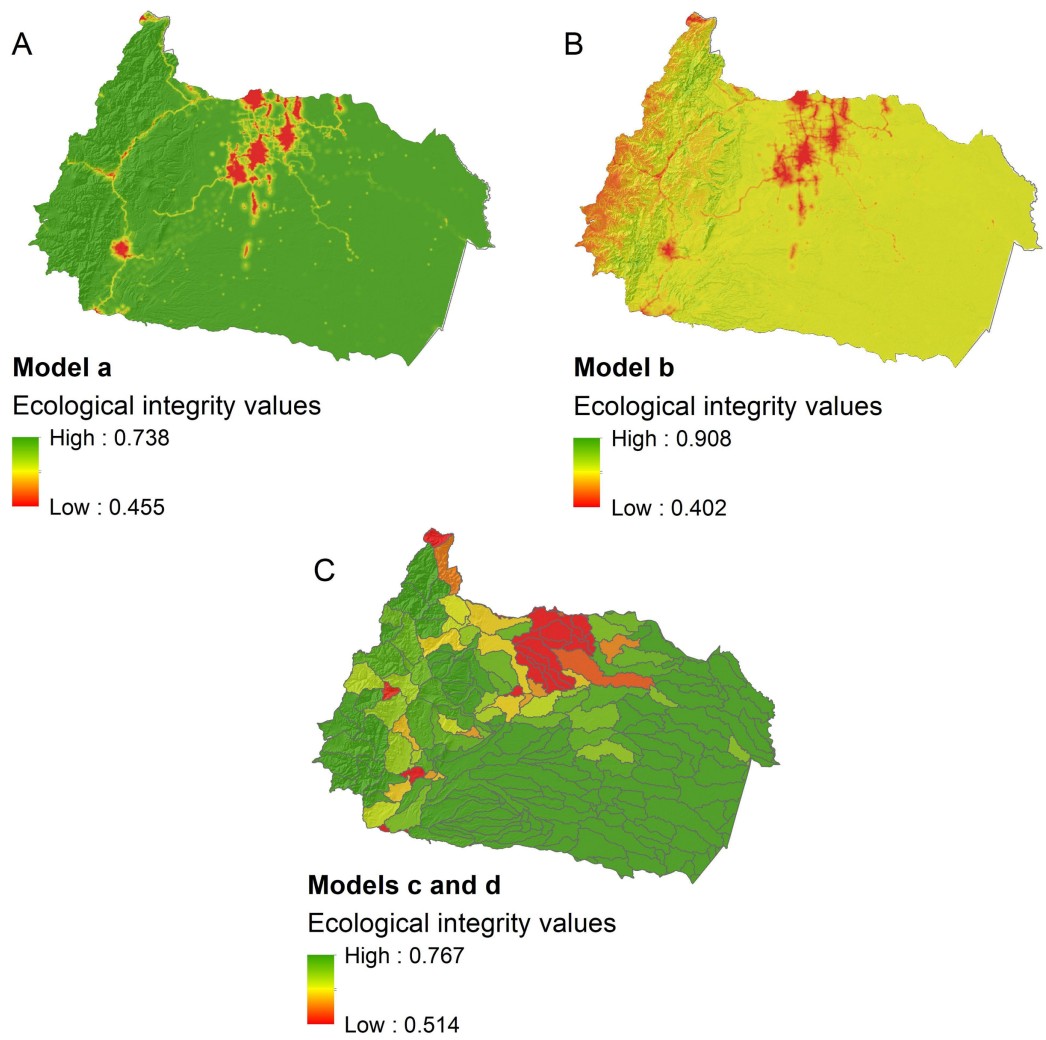

**Figure 4 Predicted ecological integrity for freshwater systems in the upper Napo river basin.** Maps of predicted ecological integrity are results of different modeling scenarios: (A) prediction of local ecological integrity from anthropogenic threat variables alone, (B) prediction of local ecological integrity from anthropogenic threat and environmental variables, (C) prediction of microbasin ecological integrity from anthropogenic threat variables alone, and from anthropogenic threat and environmental variables.

maps are assumed to be a good proxy of the condition of freshwater ecosystems (*Linke, Turak & Nel, 2011*), they are used in conservation planning at regional scales that may involve prioritization for protection of rivers with a good ecological condition over disturbed and expensive rivers to manage or restore (*Neeson et al., 2016*; *Linke, Hermoso & Januchowski-Hartley, 2019*). Nevertheless, the success of such planning depends on how effective threat maps are as a surrogate of the ecological condition of rivers. From this perspective, our study is the first to analyze spatial information on anthropogenic threats to freshwater ecosystems and its relationship with ecological integrity across an Andean–Amazon basin. With this contribution, we validate an approach (i.e., threat maps generated by GIS) to inform conservation planning at a

regional scale in the upper Napo basin and in other basins of the region that face similar environmental challenges such as the Marañon, Ucayali, and Putumayo basins.

## How well do anthropogenic threat maps predict the ecological integrity?

Our study found that the ecological integrity of rivers and microbasins of the upper Napo basin tends to decrease with increasing anthropogenic threats. Although our regression models showed modest power for predicting the integrity of unsampled rivers and microbasins (between $R^2$ 0.16 and $R^2$ 0.47), other studies for large and diverse regions have found a similar predictive ability of ecological integrity based on spatial data on human threats. For Amazonian low-order streams, between 10% and 35% of the variation in habitat measures were explained by maps of anthropogenic and environmental predictors (*Leal et al., 2016*). For United States watersheds, spatial data on stressors accounted for 1–12% of the variation in site-level biological and habitat variables of streams (*Thornbrugh et al., 2018*). According to these studies, the heterogeneous nature of the stream systems, multiple stressors, and different land-use histories in these regions hinder our ability to depict the relationships between maps of human threats and ecological integrity patterns (*Bücker et al., 2010*; *Leal et al., 2016*; *Schinegger et al., 2016*). Therefore, it is reasonable to find modest predictive power for the upper Napo basin models (*Thornbrugh et al., 2018*). Despite these constraints, our results confirm that direct ecological-integrity measures of rivers in this basin are tied to anthropogenic pressures that were independently derived from spatial data. This finding is especially encouraging since both indices (EII and threats maps) represent a drastic reduction in information content. Therefore, our study supports the use of threat maps to provide a coarse representation of actual conditions of rivers throughout the upper Napo basin, with explicit acknowledgment of the associated uncertainty.

## Which anthropogenic threats and environmental variables best explain the ecological integrity of rivers?

Our study identified *roads* as a relevant predictor of river's integrity in the upper Napo River basin. The specific impact of roads to rivers has been considered to be low when compared to more detrimental human activities, such as agriculture or mining (*Cooper, 2011*; *Esselman & Allan, 2011*). However, as shown in the threat maps, human settlements, agricultural activities, water consumption, and mining appear clustered near roads (Fig. 3). This pattern suggests that roads are associated with the presence of other threats to ecological integrity, indirectly linking roads to integrity. Given the rough and steep terrain that dominates large portions of this basin, the roads that can be established tend to run through the more accessible areas, thus concentrating the spatial distribution of other threats such as agriculture (*Suárez et al., 2012*).

According to our models, *human settlements* and *oil activities* are also relevant predictors of river's ecological integrity. In the Amazon basin, cities and villages are associated with high deforestation rates, which alter the habitat integrity of rivers. Human settlements also increase erosion, runoff of waters with contaminants (oil, heavy metals,

pesticides, sediment, nutrients, pathogens), and discharges of untreated effluents from domestic and industrial activities to nearby rivers (*Celi, 2005*; *Alho, Reis & Aquino, 2015*). Oil-related activities in the Andean-Amazon basins have received enormous attention due to the frequent cases of malpractice and oil spills that affect the health of rivers (*Fontaine, 2003*). Moreover, oil activities in the Andean-Amazon region have historically encouraged the proliferation of other human pressures, such as roads and agricultural expansion (*Suárez et al., 2009*, *2012*).

The strength of our predictive models increased significantly when environmental variables were mapped and used to estimate ecological integrity with anthropogenic threats. Specifically, sites with greater *slopes* have better ecological integrity, which results from two combined factors. First, steep slopes lead to rivers with riffles and rapids, allowing them to regain healthier ecosystem qualities in areas affected by human activity (*Rosgen, 1994*; *Newbury, 2013*). Second, steep areas may constrain the expansion of human economic activities, whereas resources that are economically-valuable, such as oil reserves and oil palm plantations, are found in the Amazonian floodplains (*Celi, 2005*). A negative relationship between *elevation* and ecological integrity was also found in our models, which is explained by the large number of remote lowland areas in the basin that have a low presence of threats.

## How do predictions differ across spatial scales?

Including environmental variables in the models improved their explanatory power, but a considerable part of the variation in ecological integrity at a local scale remains unexplained. As expected, ecological integrity measured in situ varied greatly among nearby rivers because of site-specific, short-term changes (e.g., flooding events), and localized human-related changes (e.g., riparian vegetation loss, overfishing) (*Allan, Erickson & Fay, 1997*; *Arthington et al., 2004*). In contrast, anthropogenic threat data do not reflect the most recent and local changes. Thus, the predictive capacity of models improved when the unit of analysis was transformed to microbasins (regional scale), and the spatial scale of both, response and predictor variables were the same. Moreover, processes such as sediment load, nutrient supply, hydrology, and channel characteristics are influenced by the landscape and upstream conditions, which were better captured by analyses at a regional scale rather than local scale (*Allan, Erickson & Fay, 1997*; *Arthington et al., 2004*).

## Implications for freshwater conservation planning and limitations of our study

We found that anthropogenic threat maps can provide rapid and cost-effective assessments of the regional spatial pattern of river's ecological integrity. These results suggest that threat maps are a valid resource for coarse-filter conservation planning in the upper Napo basin and other Andean-Amazon basins where measures of the condition of rivers are absent or prohibitively expensive to collect. We also found this mapping approach is especially effective at depicting the impacts of human activities that facilitate the incidence of other threats. For example, the effect of roads, settlements, and oil

activities, which were significant predictors in our model, is probably mediated through the combined influence of several human activities associated with these predictors.

The use of anthropogenic threat maps for freshwater conservation planning has limitations that are important to acknowledge. First, anthropogenic threat maps have different caveats according to the spatial scale of the conservation actions. Since the explanatory power of the models at a local scale is limited, it would always be advisable to have a finer scale in situ measurements of ecological integrity for conservation or restoration projects focused on specific target rivers. Models at the microbasin (regional) scale, although more explanatory, do not offer estimates of the ecological integrity at specific rivers within basins. Therefore, reliable use of anthropogenic threat maps is, for example: prioritizing which un-sampled micro-watershed require targeted assessments, searching for reference rivers in areas were high ecological integrity is predicted, or identifying sites that will need monitoring programs because of the expected increase in oil, urban, and road development in the basin.

Second, the ability of anthropogenic threat maps to explain the integrity of rivers can be influenced by the indicators used to assess the condition of the river. For example, in our study, ecological integrity was not inferred by hydroelectric power plants, despite of their growing and significant impact on connectivity of Andean–Amazon rivers (*Winemiller et al., 2016*; *Anderson et al., 2018*). However, this impact is localized for small invertebrates with limited dispersal capabilities, but massive for vertebrates that depend on the longitudinal connectivity provided by rivers. Therefore, management initiatives that favor this mapping approach must combine assessments of other threats that consider longitudinal, lateral, and temporal connectivity in the watershed (*Winemiller et al., 2016*; *Anderson et al., 2018*). Another example of the relevance of the indicators of ecological integrity was the limited role of the oil activities that we found in our modeling. Such result could be explained by the fact that our EII does not consider hydrocarbon, metallic elements, and other oil-related chemicals in the water, which might constrain the sensitivity of our index.

Finally, there is also room for improving investigations on the link between threat maps and the condition of Andean–Amazon rivers. Even though our study was based on extensive fieldwork, sample sites were not always evenly distributed along the whole gradient of stressors, such as aquaculture farms and hydroelectric, which might explain the week response of ecological integrity to these stressors (*Schinegger et al., 2016*). Improving the sampling to better represent the gradient and different combinations of stressors would help disentangle single and joint effects (*Schinegger et al., 2016*). In addition, our study assumed a simple scenario where the effects of threats to the ecological integrity were a negative linear response (*Thornbrugh et al., 2018*). However, an improved sampling could additionally help to elucidate if the relationships between stressors and ecological integrity better fit a non-linear function.

## CONCLUSIONS

Can we trust on anthropogenic threat maps as surrogates of the ecological integrity of rivers? This is a central question for freshwater conservation planners, who usually lack

resources for carrying out extensive field surveys along landscapes. In this context, our study found the prediction accuracy of ecological integrity is reasonable high when environmental factors are added to the threat maps as predictors, and when the analysis are carried out at regional scales. These results have broad applications for planners, which could make use of such maps as a first step for coarse prioritizations of conservation and management actions. However, for further actions at finer scales, target rivers should be subjected to more detailed evaluations in order to have site-level measurements of the integrity and local stressors. Thus, the knowledge gained by this study offers insights into the usefulness and limitations of anthropogenic threat maps and helps with the prioritization of conservation and management actions at a regional scale in the Andean–Amazon basins.

## ACKNOWLEDGEMENTS

Data gathering and processing was a collaboration of members of the Aquatic Ecology Laboratory at Universidad San Francisco de Quito and Universidad Tecnológica Indoamérica. We thank the Cayambe-Coca Ecological Reserve and the village of Oyacachi for access to sites; the people of Oyacachi for field assistance. We also thank Maja Celinscak, Eduardo Toral, Nathy Quiroz, Luis Granizo, José Schreckinger, Rommel Arboleda, Jaime Culebras, and Ítalo Tapia for the field assistance. The following institutions of Ecuador provided spatial information on anthropogenic threats in the Napo basin: The Nature Conservancy, Instituto Nacional de Estadística y Censos, Agencia de Regulación y Control Minero, Agencia de Regulación y Control de Electricidad, Programa Socio Bosque and Secretaria del Agua. We are very grateful to all people and institutions mentioned above.

### Funding

This study was funded by Partnership for Enhanced Engagement in Research (PEER) Program from USAID and NSF Collaborative Dimensions of Biodiversity grant (awards: DEB-1046408, DEB-1045960, and DEB-1045991), Universidad Tecnológica Indoamérica ("Evaluación química y biológica de la calidad de agua de la cuenca del Río Napo, Ecuador"; Q2012-10 to Juan M. Guayasamin), and collaboration grant USFQ ("Recursos de agua dulce y biodiversidad en la Cuenca del Napo" to Andrea C. Encalada). The funders had no role in study design, data collection and analysis, decision to publish, or preparation of the manuscript.

### Grant Disclosures

The following grant information was disclosed by the authors:
Enhanced Engagement in Research (PEER) Program from USAID and NSF Collaborative Dimensions of Biodiversity Grant Awards: DEB-1046408, DEB-1045960, and DEB-1045991.

Universidad Tecnológica Indoamérica: ("Evaluación química y biológica de la calidad de agua de la cuenca del Río Napo, Ecuador"; Q2012-10 to Juan M. Guayasamin), and collaboration grant USFQ ("Recursos de agua dulce y biodiversidad en la Cuenca del Napo" to Andrea C. Encalada).

## Competing Interests

The authors declare that they have no competing interests.

## Author Contributions

- Janeth Lessmann performed the experiments, analyzed the data, contributed reagents/materials/analysis tools, prepared figures and/or tables, authored or reviewed drafts of the paper, approved the final draft.
- Maria J. Troya performed the experiments, analyzed the data, contributed reagents/materials/analysis tools, prepared figures and/or tables, authored or reviewed drafts of the paper, approved the final draft.
- Alexander S. Flecker conceived and designed the experiments, contributed reagents/materials/analysis tools, authored or reviewed drafts of the paper, approved the final draft.
- W. Chris Funk conceived and designed the experiments, contributed reagents/materials/analysis tools, authored or reviewed drafts of the paper, approved the final draft.
- Juan M. Guayasamin conceived and designed the experiments, performed the experiments, analyzed the data, contributed reagents/materials/analysis tools, authored or reviewed drafts of the paper, approved the final draft.
- Valeria Ochoa-Herrera performed the experiments, analyzed the data, contributed reagents/materials/analysis tools, authored or reviewed drafts of the paper, approved the final draft.
- N. LeRoy Poff conceived and designed the experiments, contributed reagents/materials/analysis tools, authored or reviewed drafts of the paper, approved the final draft.
- Esteban Suárez conceived and designed the experiments, contributed reagents/materials/analysis tools, authored or reviewed drafts of the paper, approved the final draft.
- Andrea C. Encalada conceived and designed the experiments, performed the experiments, analyzed the data, contributed reagents/materials/analysis tools, authored or reviewed drafts of the paper, approved the final draft.

## Field Study Permissions

The following information was supplied relating to field study approvals (i.e., approving body and any reference numbers):

The Ecuadorian Ministry of Environment approved this study (Permits: #56-IC-FAU/FLO-DPN/MA, MAE-DNB-CM-2015-0017).

## Data Availability

Data is available in Appendix C, which has the information used to build the models. Specifically, it presents a description of each sample site in the Napo Basin: its geographic

coordinates, its ecological integrity index and the values of the environmental and threat variables.

## Supplemental Information

Supplemental information for this article can be found online at http://dx.doi.org/10.7717/peerj.8060#supplemental-information.

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
