# Peer review of "Validating anthropogenic threat maps as a tool for assessing river ecological integrity in Andean–Amazon basins"

_PeerJ, doi:10.7717/peerj.8060_

## Round 0.1 · original submission · Major Revisions

I believe that both reviewers raised a number of important issues that you will want to address in your revision. Please provide a point-by-point response to each together with your revised manuscript.

Reviewer 1 ·

Basic reporting

The manuscript aims to validate anthropogenic threat maps as a tool for assessing river ecological integrity in Andean-Amazon basins using distinct statistical methods, e.g. GLM, cross-validation. In general, the manuscript is well written using clear and professional English, and the literature review and background introduction are sufficient. Also, the structure, figures, tables are okey. However, I have two main concerns here:
1) Method: more details are needed. For instance, which functions are used for different R packages. It would be great if the authors can share the raw data and R scripts as an appendix, so that the results can be re-produced for other catchments.

2) Calculation of ecological integrity: the authors gave different weight to various categories, but the references are lack. Please add them (line 215).

3) Validation of the models: do you have a criterion for a reliable or satisfactory validation (you used cross-validation)? I am asking because your prediction powers are very low (Table 2). In addition, please try to compare your results with relevant or similar studies.

Experimental design

no comment. Regarding the method, please see above.

Validity of the findings

The idea (i.e., anthropogenic threat maps) is novel and creative. However, the results in particular prediction powers are not strong enough. Please see my comments above.

Additional comments

NA

Reviewer 2 ·

Basic reporting

The paper is handling a very important and urgent topic, it is very well and in most sections clearly written. It is structured in a very professional way and I really enjoyed reading it.

However, clear hypotheses are missing at the end of the introduction and I would love to see them stated in a revision of this article.

Experimental design

This article represents primary research within Aims and Scope of the journal, and they should clearly be published within, after a more major than minor revision.

As stated above, the hypotheses and the related research questions are not defined clearly, this needs to be done in a revision of the article. It is somehow clear, how this research fills the identified knowledge gap, but I would love to see a broader discussion of this, backed up with more international citations on this field of conservation planning.

Basically, the investigation seems to be performed at high technical & ethical standard, in terms of methods and results, I however have a few very specific comments:

Materials & Methods
Lines 151/152: Please give a very short explanation of the páramo ecosystems, as readers not familiar with the Andean/Amazonian system might not recognize this ecosystem type.

Lines 161-164: Please proof how the independence between sites was considered in the sampling design and give a reference.

Line 194: Not clear what is meant by the "microbasin scale", please define and explain already at this stage. Also, not clear to me how and especially why the measured values were compared to the modeled ones - please explain.

Lines 197-202: By just averaging the three single indices, the computation of the Ecological Integrity Index (EII) is done in very simple an generalizing way, which often underestimates the effect/impact of single categories. Previous studies as e.g. Chen et al. (2017)
Chen, K., Hughes, R. M., Brito, J. G., Leal, C. G., Leitão, R. P., de Oliveira-Júnior, J. M., ... & Hamada, N. (2017). A multi-assemblage, multi-metric biological condition index for eastern Amazonia streams. Ecological indicators, 78, 48-61.

or Schinegger et al. (2012)
Schinegger, R., Trautwein, C., Melcher, A., & Schmutz, S. (2012). Multiple human pressures and their spatial patterns in E uropean running waters. Water and Environment Journal, 26(2), 261-273.

have shown more complex ways how to calculate stressor indices or MMI's in a more complex way and I would love to see a more complex EII calculation as scenario B, in order to compare the results (e.g. a weighted index). If this is not possible, I at least would expect a comprehensive discussion of the limitations of such an approach and a justification (with references) why this one was chosen/sticked with.

Lines 205-2013: Please highlight the sources of datasets here, as done later on for environmental variables. Also, I'd like to see more details, as e.g. resolution of datasets etc.in the Appendix table B.

Lines 2015-2016: Not clear to me, where the specialized references can be found, information in Appendix B is quite short and should be extended, also for the references (full dataset reference and publication citation, if possible.)

Lines 2019-220: According to which reference did you select the maximum distance/buffer? Please provide scientific proof with references and/or a justification why there is no references.

Lines 233-234: As stated before, the term "microbasin" and the spatial framework for the whole study is not clearly explained/defined at all, this needs to be done, maybe earlier in the manuscript already, when it is mentioned the first time.

Lines 247-257: If possible in this journal, give an example of the GLM calculation in the way of a formula with the darious elements.

Line 253: Please give a reference for your AIC selection procedure.

Line 258-270: Please justify why you used this validation procedure and give a reference for it.

Results
Line 290: The "river scale" is not defined/mentioned before, please do this and include it in your spatial framework description.

Validity of the findings

Impact and novelty of findings is assessed, however, I miss a bit a discussion in a broader context, e.g. in terms of strategic conservation planning with related cost, e.g. the works of Simone Langhans and especially Virgilio Hermoso, which would fit very well into the discussion section.

Also, I would like to see a "Uncertainties & Limitations" paragraph in the discussion, where weaknesses of the study's approach are discussed, e.g. limitation of macroinvertebrates to indicate certain large-scale stressors (or the ability of other indicators to do better), and any other issues, especially related to the large spatial scale and related biases in datasets etc. --> here, you could read a bit the literature, e.g. Dave Allan et al. (2004 --> Landscape scale studies). Also, check the literature of Bob Hughes and his conclusions/limitations for South American IBI's.
In terms of fragmentation, I also propose to discuss the paper of Anderson et al. (2018) in a wider context, as the connectivity aspect, which is very crucial for the investigated area, is not well covered in the selected stressor-datasets.
Anderson, E. P., Jenkins, C. N., Heilpern, S., Maldonado-Ocampo, J. A., Carvajal-Vallejos, F. M., Encalada, A. C., ... & Salcedo, N. (2018). Fragmentation of Andes-to-Amazon connectivity by hydropower dams. Science advances, 4(1), eaao1642.

Finally, in the conclusions, I also see a few speculations which should clearly be identified as such, e.g. that the findings assure reliable results, which is - in such a generalized way - simply not true.

Additional comments

Finally, however, I think this study is really important and good work, which should be published in this journal, after revision.

---

## Round 0.2 · accepted · Accept

I believe you have addressed both reviewers' comments thoroughly and that the revised manuscript is much stronger as a result.